# The economic burden of low back pain in KwaZulu-Natal, South Africa: A prevalence-based cost-of-illness analysis from the healthcare provider's perspective

**Morris Kahere**[1]*, **Cebisile Ngcamphalala**[2], **Ellinor Östensson**[3,4], **Themba Ginindza**[1,2]

**1** Discipline of Public Health Medicine, School of Nursing and Public Health, University of KwaZulu-Natal, Durban, South Africa, **2** Cancer & Infectious Diseases Epidemiology Research Unit (CIDERU), College of Health Sciences, University of KwaZulu-Natal, Durban, South Africa, **3** Department of Women's and Children's Health, Karolinska Institute, Stockholm, Sweden, **4** Department of Medical Epidemiology and Biostatistics, Karolinska Institutet, Stockholm, Sweden

* mrrskhr@gmail.com

**Data Availability Statement:** Data from this study are the property of the Government of South Africa and University of KwaZulu-Natal and cannot be

## Abstract

### Background

Low back pain (LBP) is a multifactorial and the most prevalent musculoskeletal disorder, whose economic burden is of global concern. Evidence suggests that the burden of LBP in increasing and will continue rising with the greatest burden occurring in low-and-middle-income-countries (LMICs). This study sought to determine the economic burden of LBP in KwaZulu-Natal, South Africa from the providers perspective.

### Methods

We used a retrospective prevalence-based cost-of-illness methodology to estimate the direct medical cost of LBP. Direct medical costs constituted costs associated with health-care utilisation in inpatient care, outpatient care, investigations, consultations, and cost of auxiliary devices. We used diagnostic-specific data obtained from hospital clinical reports. All identifiable direct medical costs were estimated using a top-down approach for costs associated with healthcare and a bottom-up approach for costs associated with inpatient and outpatient care.

### Results

The prevalence of chronic low back pain CLBP was 24.3% (95% CI: 23.5–25.1). The total annual average direct medical costs associated with LBP was US$5.4 million. Acute low back pain (ALBP) and CLBP contributed 17% (US$0.92 million) and 83% (US$4.48 million) of the total cost, respectively. The per patient total annual average direct medical cost for ALBP and CLBP were US$99.43 and US$1,516.67, respectively. The outpatient care costs contributed the largest share (38.9%, US$2.10 million) of the total annual average direct medical cost, 54.9% (US$1.15 million) of which was attributed to nonsteroidal-anti-

made publicly available. All interested readers can access the data set from the Chairperson of the South Africa Health Research and Ethics Committee and University of KwaZulu-Natal Biomedical Research Ethics Committee (BREC) from the following contacts: The Chairperson of South Africa Health Research and Ethics Committee, email: hrkm@kznhealth.co.za, Tel: +27 (033) 395 2805. The Chairperson BIOMEDICAL RESEARCH ETHICS ADMINISTRATION Research Office, Westville Campus, Govan Mbeki Building University of KwaZulu-Natal P/Bag X54001, Durban, 4000 KwaZulu-Natal, South Africa Tel.: +27 31 260 4769 Fax: +27 31 260 4609 Email: BREC@ukzn.ac.za.

**Funding:** This study was funded by the University of KwaZulu-Natal College of Health Sciences (CHS Scholarship). The funders had no role in study design, data collection and analysis, decision to publish, or preparation of the manuscript.

**Competing interests:** The authors have declared that no competing interests exist.

inflammatory drugs (NSAIDs). The total average cost of diagnostic investigations was estimated at US$831,595.40, which formed 15.4% of the average total cost.

## Conclusion

The economic burden of LBP is high in South Africa. Majority of costs were attributed to CLBP. The outpatient care costs contributed the largest share percent of the total cost. Pain medication was the main intervention strategy, contributing more than half of the total outpatient costs. Measures should be taken to ensure guideline adherence. Focus should also be placed towards development of prevention measures to minimise the cost.

## Background

Low back pain (LBP) is a global public health problem that occurs in high-income-countries (HICs) and low-and-middle-income countries (LMICs) across all age groups [1]. Despite the technological advancements in diagnosis and the advent of several intervention approaches in the recent years, the mosaic of the pathophysiology of LBP is still far from being understood. Thus, LBP is still known to cause significant socio-economic burden to the society [2]. According to the Global Burden of Disease (GBD) 2017, the years lived with disability (YLD) due to LBP has increased by 52.7%, from 42.5 million in 1990 to 64.9 million in 2017 [1, 3]. Globally, LBP is now the leading cause of disability [4]. The burden attributed to LBP is predicted to continue increasing, particularly in LMICs where there is limited health coverage and pro-communicable disease control [1]. Great strides should be taken to address this increasing burden and to alleviate the impact it is imposing on health and socio-economic systems. The magnitude of the burden of LBP can be expressed in prevalence, incidence, and cost estimates.

The prevalence and incidence estimate of LBP vary among studies due to differences in definitions of LBP and methodologies used in different studies over time. Additionally, Anema et al. reported that these variations are also influenced by the differences in the healthcare seeking behavior, local socio-cultural systems and beliefs around cause and effect [5]. This lack of coherence and homogeneity makes it difficult to compare different studies. However, in the Western world, the point prevalence of LBP has been reported to be 15–30%, with an estimated 1-month prevalence of 19–43% and a lifetime prevalence of up to 85% [1]. A systematic review by Morris et al. investigating the prevalence of LBP in Africa showed a pooled lifetime, 12-months and point prevalence of LBP of 47%, 57% and 39% respectively [6]. These high prevalence estimates observed in the western world can be attributed to a great awareness of LBP and the willingness to report symptoms as compared to other parts of the world [1, 7].

About 90% of LBP cases are not severe and normally resolves within a few days to a few weeks but up to 10% of cases will develop into chronic low back pain (CLBP). According to Watson et al. (2010) most patients do not make a full recovery but will have "flare-ups" against a background of CLBP, meaning that the majority of patients will have recurrent symptoms [8]. Regardless of the small percentage of CLBP sufferers, this group is responsible for the majority of the economic burden incurred [9]. A USA study of insurance claims by Hashemi et al. showed that up to 8.8% of LBP sufferers had symptoms that lasted for a year and accounted for up to 84.7% of the total costs attributed to LBP [10]. Similarly, in a study of the UK working population, only 3% of the patients had symptoms lasting for more than three months but contributed to 33% of the benefits paid out during the period of that study [8, 11].

A cross-sectional study by Ekman et al. investigating the burden of CLBP in Sweden reported that the total annual direct and indirect cost of CLBP per patient were estimated at US$2 900 and US$16 600 in 2002 prices, respectively [9]. Another cross-sectional Switzerland study by Wieser et al. reported the direct costs of CLBP to be €2.3 billion and indirect costs were estimated at €4.1 billion using the human capital approach and €2.2 billion using the friction cost method, representing 2.3% of the total gross domestic production [12]. Walker et al. estimated the direct cost of LBP at AU$1.02 billion and indirect cost at AU$8.15 billion among the Australian adults [13]. In the Netherlands, van Tulder et al. reported that the total annual direct costs of LBP were estimated at US$367.6 million, while the total annual indirect costs were estimated at US$4.6 billion [14]. Estimates of the economic burden of LBP in the United States, for both direct and indirect costs, range from $84.1 billion to $624.8 billion [2].

Fianyo et al. [18] investigated the cost of LBP and lumbar radiculopathy in Lomé. This was the only cost-of-illness study found in Africa after an extensive search of literature. Fianyo et al. reported that, the average total cost for LBP in hospital consultations was estimated at US $107.2 (range: US $ 5.8 and US $ 726.1). This cost constituted the direct cost which were US $56.3 representing about 53% of the total cost and indirect cost of US$50.3 which was 47% of the total costs incurred. Of the direct costs, 36.9% were direct medical costs and 16.1% were direct non-medical costs. About 68.9% (71) of the participants reported that their budget was stretched by the costs of low back pain management some of which ending up in debt. Only 13 patients reported that their medical care costs were catered for by their employers. About 87.1% (27) were getting familial financial support with an average of US$27.5 cash donations per patient. Only one patient underwent a surgical procedure which costed US$1600 but 15 participants had been offered surgery. On the other hand, the intangible costs were largely determined by discomfort in everyday life and discomfort in emotional life.

The national development plan (NDP) and health policy in South Africa, seek to decrease the prevalence of non-communicable diseases and improve health outcomes. Plans are also underway to implement the national health insurance (NHI) to ensure accessibility to health and promote quality in health. As the leading driver of disability, understanding of the costs associated with LBP remains critical to inform health care policy decisions and subsequently improve management of LBP. Using patient health records from five hospitals this study sought to close that knowledge gap by estimating the economic burden of LBP among adults (aged $\geq$ 18 years), by estimating direct medical costs including inpatient- and outpatient care for management of LBP in tertiary care.

## Materials and methods

### Study area

This was a prevalence-based cost of illness study conducted in five randomly selected provincial public hospitals in the eThekwini district of KwaZulu-Natal in South Africa. KwaZulu-Natal is an East coastal province with the second largest population in South Africa. The 2019 population and housing census estimated the population of KwaZulu-Natal to be approximately 11.3 million people (19.2% of the total population) [15]. The KZN GDP per capita is estimated at US$10 406, which makes it fall in the low-income category [16].

### Study setting

This was a hospital-based study which included five primary public hospitals (viz; Addington Hospital, Mahatma Gandhi Memorial Hospital, Prince Mshiyeni Memorial Hospital, Hillcrest Hospital, and Clairwood Hospital) in the eThekwini health district of KwaZulu-Natal. Addington is a district and regional hospital with 471 beds and 2200 employees. Addington hospital

offer a variety of services including inpatient occupational therapy services for disabled patients. Mahatma Gandhi is a 350 bedded hospital offering inpatient and outpatient care services, including inpatient physiotherapy and occupational therapy services for musculoskeletal patients. Prince Mshiyeni is a 1075 bedded hospital located in Umlazi township. Prince Mshiyeni offers both district and regional services and a variety of clinics available within. Hillcrest hospital is a 167 bedded specialised chronic pain patients' hospital that takes patients who need nursing care. These patients are referred from the hospitals throughout the entire province of KwaZulu-Natal. This hospital also offers outpatient services for chronic medication and rehabilitation. Clairwood is a 275-bedded specialised rehabilitation and convalescent hospital located in the township of Clairwood. A simple random sampling technique, using the hat method, was used to select the participating hospitals.

## Method of costing

From a healthcare provider's perspective, we employed a prevalence-based method [17] with a bottom-up approach identifying related cost procedures and activities to estimate direct medical costs associated with outpatient and inpatient hospital care for LBP in KwaZulu-Natal between 2018–2019 [17]. Patient records from five eThekwini District Hospitals were accessed by the trained research assistants to determine the number of patients diagnosed with LBP. Low back pain diagnoses classified as per the international classification of diseases (ICD)-10 codes, specifically M40 –M54, M96 and M99 with subclassification codes related to the lumbar spine or lumbosacral spine were included in this study. Codes which did not allow practical delineation of the lumbar spine were excluded [18].

## Management of LBP in South Africa

The management of LBP in South Africa follows a referral pattern. With the health care system organised into four hierarchical levels of access. The South African primary health care is comprised of primary health care centres (PHC), community health care centres (CHC), local clinics and general practitioners (GP). The PHC is the first step in the provision of health care for LBP patients and can only dispense pain medication. Patients requiring further investigations will be referred to district hospitals where general support in diagnostics (laboratory tests and imaging studies), treatment, care, counselling, and rehabilitation is provided. The treatment of LBP at a district hospital primarily involves prescription of pain medications, mainly non-steroidal anti-inflammatory drugs (diclofenac and celecoxib), opioids (tramadol) and antidepressants (amitriptyline) [19]. Tramadol and a combination of tramadol and diclofenac are the most prescribed medication. Antidepressants are prescribed if a patient presents with symptoms of depression, and they can also be used in combination with NSAIDs and or Tramadol. Antidepressants are often used primarily to treat pain (often at a lower dose than for depression). Anticonvulsants are also commonly used in this way. Physiotherapeutic management involves exercises and stretches, transcutaneous electrical nerve stimulation, ultrasound, and laser therapy. Foot orthotics (insoles) are prescribed if the biomechanics of the foot is suspected as the root of the problem. Invasive procedures can be recommended in cases where all other ono-invasive options have been exhausted. The flow of events is depicted in Fig 1.

## Costs

We collected all identifiable direct medical costs incurred due to consultations, resource utilisations for inpatient and outpatient care events associated with LBP diagnosis/special investigations and treatment (including medications, any invasive procedures, rehabilitation, use of any auxiliary devices) [20]. We computed all costs at the 2019 price level and converted the

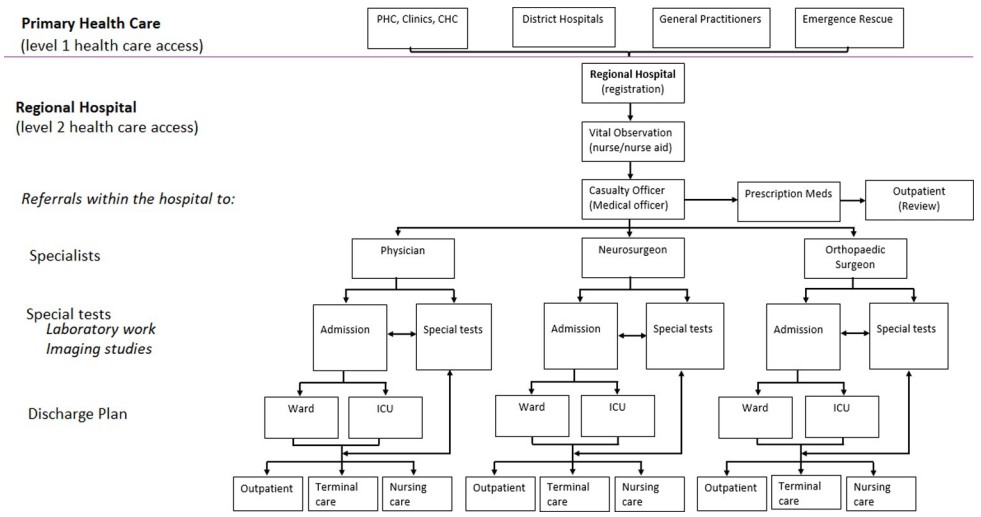

**Fig 1. Patient care pathway.**

currency from the South African Rand (ZAR) to United States Dollar ($) using the 2019 average exchange rate (US$1 = ZAR 14.45).

## Direct medical costs

To estimate the total direct medical costs associated with LBP, we estimated the average cost of each care event documented in the hospital patient records. The average cost for each care component was multiplied by the total corresponding number of patients identified in that component. All cost-generating events were identified and attributed a monetary value based on market or private sources obtained through consultation with senior medical practitioners from private sector (Joint Medical Holdings Ltd).

## Ethical considerations

The study was approved by University of KwaZulu-Natal Biomedical Research Ethics Committee (BREC) (Ref No: BREC/00000205/2019) and the KwaZulu-Natal Department of Health Ethics (Ref No: KZ_201909_002). Gatekeeper permissions were sort from participating institutions prior to the commencement of data collection. To guarantee the anonymity of each participant, the names of respondents, their addresses or other identifying information were included in the questionnaires, but rather each participant was assigned a study ID which was only accessed by the researcher. There was no human participation in this study, as it was a retrospective study of hospital health records for low back pain patients who presented to the hospitals between 2018 and 2019, therefore, no participants consent was required.

## Results

### Participants

A total of 12169 files were retrieved. The prevalence of CLBP was 24.3% (2957/12169). Women represented 55.2% (n = 6716) and Men 44.8% (n = 5453) of the study population, Table 1. The mean ± standard deviation age was 57.6±15.2 years. Notably, young adults (aged 18–27 years) represented the smallest percentage of the study population 5.2% (n = 636).

**Table 1. Demographic characteristics.**

| Age (years) Mean±SD = 57.6±15.2 | Women (n = 6716) | | Men (n = 5453) | | Overall (N = 12169) | | |
|---|---|---|---|---|---|---|---|
| | (n) | (%) | (n) | (%) | (n) | (%) | 95% CI |
| 18–27 | 387 | 5.76 | 249 | 4.57 | 636 | 5.23 | 4.84–5.64 |
| 28–37 | 924 | 13.76 | 522 | 9.57 | 1446 | 11.88 | 11.31–12.47 |
| 38–47 | 1797 | 26.76 | 1371 | 25.14 | 3168 | 26.03 | 25.26–26.82 |
| 48–57 | 1975 | 29.41 | 1675 | 30.72 | 3650 | 29.99 | 29.18–30.82 |
| 58–67 | 1041 | 15.50 | 1068 | 19.59 | 2109 | 17.33 | 16.66–18.02 |
| 68+ | 592 | 8.81 | 568 | 10.42 | 1160 | 9.53 | 9.02–10.07 |

Note: SD = Standard Deviation

Whilst a majority of the population were Women, the proportion of Men were higher in the age category, 58–67 (1068 Men compared to 1041 Women).

## Direct medical costs associated with outpatient care of ALBP

We performed an age-and-gender stratified costing analysis, **Table 2**. The estimated average direct medical cost associated with outpatient care for ALBP increased with increasing age for both genders, Table 2. The total annual average direct medical costs for ALBP were estimated at US$915,948.87 whilst the cost per patient was estimated at US$99.43. The main cost drivers for ALBP were pain medication consisting of opioids (tramadol) and non-steroidal anti-inflammatory drugs (NSAIDs) mainly diclofenac and celecoxib which accounted for 83% (US$760,294.08) of the total cost, and per patient cost of US$82.53. NSAIDs, opioids and rehabilitation accounted for 68.5% (US$626,939.04), US$68.06 per patient, 14.6% (US$133,355.04), US$14.48 per patient and 5.4% (US$49,138.92), US$5.33 per patient for ALBP respectively.

## Direct medical costs associated with chronic low back pain

The total annual average direct medical cost for CLBP was estimated at US$4,48 million with the costs per patient estimated at an annual average cost of US$1,516.67. The highest average cost per patient was observed among the elderly population in both genders, female (US$2,219.59) and male (US$1,932.33), **Table 3**. As per the cost variables, inpatient care contributed the highest cost constituting 46.31% (US$2.08 million) of the total annual average costs for CLBP followed by outpatient care 26.5% (US$1.19 million), investigation 18.5% (US$831,595.40), specialists 7.2% (US$323,880.63) and auxiliaries 1.4% (US$62,366.84). The main driver of the outpatient costs for CLBP were medication, which contributed 79.6% (US$947,184.96) of the total outpatient costs and 21.12% of the total direct medical cost. NSAIDs were responsible for more than half 55.67 (US$527,124.00) of the total medication costs and opioids accounted for 24.3% (US$229,733.28) of the total cost of medication. Men presented with higher costs across the age groups compared female's counterpart (Fig 2). Additionally, the costs increased with age.

## Overall estimated cost for LBP

Overall, the total annual direct medical cost for LBP was estimated at US$5.4 million (**Table 4**) with costs for ALBP and CLBP accounting for 17% (US$0.92 million) and 83% (US$4.48 million) of the total cost, respectively. The per person annual total average direct medical cost for ALBP and CLBP were estimated at US$99.43 and US$1,516.67 respectively. The total costs among those aged 18–27 were estimated at US$33,281.37 with the costs more than

Table 2. Costs associated with acute low back pain (n = 9212).

| Outpatient care | 18–27 (n = 606) | | 28–37 (n = 1148) | | 38–47 (n = 2502) | | 48–57 (n = 2671) | | 58–67 (n = 1354) | | ≥ 68 (n = 931) | | Total counts (n = 9212) | Unit price (US$) | Total price (US$) |
|---|---|---|---|---|---|---|---|---|---|---|---|---|---|---|---|
| | F (n = 373) | M (n = 233) | F (n = 799) | M (n = 349) | F (n = 1492) | M (n = 1010) | F (n = 1545) | M (n = 1126) | F (n = 694) | M (n = 660) | F (n = 426) | M (n = 505) | | | |
| **Visits** | 403 | 241 | 999 | 407 | 2520 | 1469 | 3699 | 2304 | 2247 | 1917 | 1543 | 1697 | 19446 | 5.19 | 100,924.74 |
| *Medication* | | | | | | | | | | | | | | *82.53* | *760,294.08* |
| **NSAIDS** | 403 | 241 | 999 | 407 | 2520 | 1469 | 3699 | 2304 | 2247 | 1917 | 1543 | 1697 | 19446 | 32.24 | 626,939.04 |
| **Opioids** | 178 | 133 | 414 | 173 | 742 | 473 | 711 | 520 | 366 | 365 | 182 | 215 | 4472 | 29.82 | 133,355.04 |
| **Rehab** | 87 | 61 | 207 | 86 | 393 | 264 | 400 | 299 | 169 | 159 | 105 | 137 | 2367 | 20.76 | 49,138.92 |
| **Insoles** | 22 | **15** | 52 | 23 | 98 | 65 | 91 | 72 | 45 | 40 | 25 | 29 | 577 | 9.69 | 5,591.13 |
| **Total cost** | 22,411.55 | **14,398.40** | 54,539.25 | 22,401.10 | 125,558.34 | 75,200.02 | 168,841.38 | 108,650.04 | 98,963.82 | 86,326.05 | 65,603.78 | 73,055.14 | | | **915,948.87** |
| **Average cost/patient** | 60.08 | **61.80** | 68.26 | 64.19 | 84.15 | 74.46 | 109.28 | 96.49 | 142.60 | 130.80 | 154.00 | 144.66 | | | *99.43* |

Table 3. Costs associated with chronic low back pain (n = 2957).

| Cost variable | Sub-category | 18–27 (n = 30) W (n = 14) | M (n = 16) | 28–37 (n = 298) W (n = 125) | M (n = 173) | 38–47 (n = 666) W (n = 305) | M (n = 361) | 48–57 (n = 979) W (n = 430) | M (n = 549) | 58–67 (n = 755) W (n = 347) | M (n = 408) | ≥ 68 (n = 229) W (n = 166) | M (n = 63) | Total counts 2957 | Unit price (US$) | Total price (US$) |
|---|---|---|---|---|---|---|---|---|---|---|---|---|---|---|---|---|
| **Inpatient** | | | | | | | | | | | | | | | *(702.33)* | *(2,076,792.28)* |
| | Ward | 27 | 38 | 52 | 119 | 398 | 444 | 449 | 822 | 482 | 621 | 383 | 169 | 4004 | 249.00 | 996,996.00 |
| | ICU | - | - | 17 | 36 | 89 | 109 | 207 | 275 | 199 | 308 | 196 | 113 | 1549 | 116.57 | 180,566.93 |
| | Nursing care | - | - | - | - | 93 | 106 | 172 | 181 | 139 | 191 | 73 | 10 | 965 | 899.68 | 868,191.20 |
| | Terminal care | - | - | - | - | - | - | 25 | 14 | 59 | 50 | 47 | - | 195 | 159.17 | 31,038.15 |
| **Outpatient** | | | | | | | | | | | | | | | *(402.49)* | *(1,190,149.62)* |
| | Total visits | 65 | 49 | 467 | 542 | 1546 | 1326 | 2190 | 2758 | 2688 | 3099 | 1236 | 384 | 16350 | 5.19 | 84,856.50 |
| | *Medication* | | | | | | | | | | | | | | *(320.31)* | *(947,184.96)* |
| | NSAIDs | 65 | 49 | 467 | 542 | 1546 | 1326 | 2190 | 2758 | 2688 | 3099 | 1236 | 384 | 16350 | 32.24 | 527,124.00 |
| | Opioids | 21 | 19 | 191 | 280 | 583 | 667 | 840 | 1064 | 1354 | 1714 | 824 | 147 | 7704 | 29.82 | 229,733.28 |
| | Antidepressants | - | - | - | 65 | 585 | 487 | 891 | 1050 | 1049 | 1290 | 472 | 223 | 6112 | 31.14 | 190,327.68 |
| | Rehabilitation | 14 | 37 | 280 | 370 | 699 | 979 | 988 | 1551 | 758 | 1222 | 399 | 319 | 7616 | 20.76 | 158,108.16 |
| **Investigations** | | | | | | | | | | | | | | | *(282.23)* | *(831,595.40)* |
| | Baseline bloods | 4 | 5 | 41 | 58 | 102 | 121 | 142 | 191 | 131 | 151 | 62 | 24 | 1032 | 150.90 | 155,728.80 |
| | X-Ray | 14 | 16 | 123 | 172 | 303 | 359 | 427 | 547 | 348 | 408 | 166 | 61 | 2944 | 67.82 | 199,662.08 |
| | MRI scan | 4 | 4 | 34 | 41 | 109 | 116 | 141 | 180 | 134 | 122 | 80 | 18 | 983 | 484.44 | 476,204.52 |
| **Specialists** | | | | | | | | | | | | | | | *(109.53)* | *(323,880.63)* |
| | Physician | 14 | 16 | 119 | 169 | 273 | 328 | 382 | 488 | 302 | 350 | 120 | 10 | 2571 | 109.35 | 281,138.85 |
| | Neurosurgeon | - | - | 3 | 4 | 16 | 21 | 27 | 33 | 26 | 31 | 27 | 27 | 215 | 110.73 | 23,806.95 |
| | Orthopaedic | - | - | 4 | 1 | 16 | 13 | 20 | 28 | 19 | 27 | 19 | 24 | 171 | 110.73 | 18,934.83 |
| **Auxiliaries** | | | | | | | | | | | | | | | *(21.09)* | *(62,366.84)* |
| | Foot insoles | 1 | 2 | 10 | 20 | 22 | 35 | 50 | 93 | 50 | 59 | 34 | 22 | 398 | 9.69 | 3,856.62 |
| | Crutches | - | - | 4 | - | 76 | 71 | 144 | 123 | 102 | 132 | 50 | 6 | 708 | 16.26 | 11,512.08 |
| | Wheelchairs | - | - | 2 | - | 16 | 24 | 29 | 58 | 37 | 59 | 25 | 9 | 259 | 181.46 | 46,998.14 |
| **Total cost ($)** | | 15104,24 | 18177,13 | 89230,26 | 131676,36 | 427620,38 | 467921,14 | 623689,59 | 828323,19 | 622199,85 | 770653,46 | 368452,49 | 121736,68 | | | **4,484,784.77** |
| **Average cost** | | 1,078.87 | 1,136.07 | 713.84 | 761.14 | 1,402.03 | 1,296.18 | 1,450.44 | 1,508.79 | 1,793.08 | 1,888.86 | 2,219.59 | 1,932.33 | | | *1,516.67* |

Note: W = women; M = men; ICU = intensive care unit

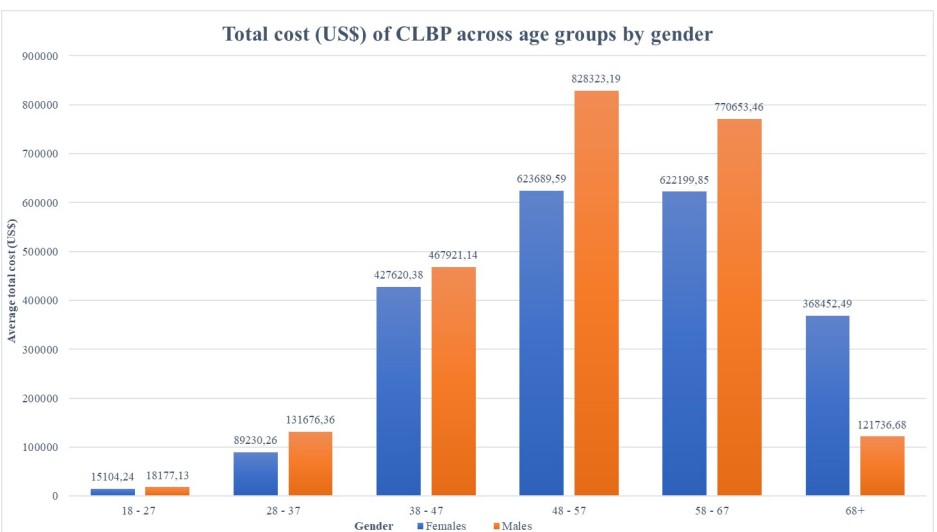

**Fig 2. Total cost of CLBP across age groups by gender.**

quadrupling for middle age groups 28–37 and 38–47 years, **Table 5**. The average total cost per patient for ALBP and CLBP is shown in **Table 6**. Outpatient care costs occurred across both ALBP and CLBP recording US$910,357.74 and US$1,190,149.62 respectively. Medication costs were comparable for both ALBP (US$760,294.08) and CLBP (US$947,184.96). Overall, the main cost driver was outpatient care which contributed 38.9% (US$2.10 million) of the total direct medical cost (US$5.4 million).

## Discussions

From the health care providers' perspective, this study estimated costs of LBP (ALBP and CLBP), in KwaZulu-Natal, South Africa. The estimated total annual direct medical cost of LBP was US$5.4 million with higher costs for CLBP compared to ALPB. There were more Women with ALBP whilst the opposite was observed with CLBP. The argument could be that due to maternal health conditions women are likely to present with ALBP whilst on the other hand this could be reflecting the general health seeking behaviour between Women and Men. General, evidence on disease pattern and health care seeking behaviour in developing worlds have consistently indicated poor health care seeking behaviour among Men compared to Women [21, 22], hence Men being likely to present with CLBP.

This study shows that CLBP was responsible for most of the cost, contributing 83% of the total cost. This concurs with a systematic review by Maetzel et al. who reported that, the small proportion of CLBP patients accounts for a large fraction of the total costs [23, 24]. This was also consistent with what was reported by Gore et al. in their study of the burden of CLBP in the United States [24]. This finding can be attributed to the fact that ALBP generally last for a few weeks requiring less visits to the hospital. Therefore the cost is mostly associated with pain medication and rehabilitation for a few weeks. On the other hand, CLBP is associated with multiple consultations for a long period of time, requiring tagerted multidiscipliney treatment approach involving multiple professionals. In some cases special investigations (laboratory and imaging studies) may be required to aid the diagnosis [3, 25–27], hence more costs. In order to reduce the burden of CLBP, a shift of focus is needed from developing guidelines for management todeveloping guidelines for prevention [28]. Future research should focus on

**Table 4. Overall estimated cost low back pain in tertiary care (N = 12169).**

| Cost variable | Sub-category | Women (n = 6716) | Men (n = 5453) | Total counts | Unit cost (US$) | Total cost (US$) | Price Source |
|---|---|---|---|---|---|---|---|
| **Inpatient care** | | | | | **170.66** | **(2,076,792.28)** | |
| | Ward | 1791 | 2213 | 4004 | 249.00 | 996,996.00 | Private Hospital |
| | ICU | 708 | 841 | 1549 | 116.57 | 180,566.93 | Private Hospital |
| | Nursing care | 477 | 488 | 965 | 899.68 | 868,191.20 | Private Hospital |
| | Terminal care | 131 | 64 | 195 | 159.17 | 31,038.15 | Private Hospital |
| **Outpatient care** | | | | | **172.61** | **(2,100,507.36)** | |
| | Total visits | 19603 | 16193 | 35796 | 5.19 | 185,781.24 | Market Price |
| | *Medication* | | | | *140.31* | *1,707,479.04* | |
| | NSAIDs | 19603 | 16193 | 35796 | 32.24 | 1,154,063.04 | Market Price |
| | Opioids | 6406 | 5770 | 12176 | 29.82 | 363,088.32 | Market Price |
| | Antidepressants | 2997 | 3115 | 6112 | 31.14 | 190,327.68 | Market Price |
| | Rehabilitation | 4499 | 5484 | 9983 | 20.76 | 207,247.08 | Market Price |
| **Investigations** | | | | | **68.33** | **(831,595.40)** | |
| | Baseline bloods | 482 | 550 | 1032 | 150.90 | 155,728.80 | Private Hospital |
| | X-Ray | 1381 | 1563 | 2944 | 67.82 | 199,662.08 | Private Hospital |
| | MRI scan | 502 | 481 | 983 | 484.44 | 476,204.52 | Private Hospital |
| **Specialists** | | | | | **26.61** | **(323,880.63)** | |
| | Physician | 1210 | 1361 | 2571 | 109.35 | 281,138.85 | Market Price |
| | Neurosurgeon | 121 | 94 | 215 | 110.73 | 23,806.95 | Market Price |
| | Orthopaedic | 102 | 69 | 171 | 110.73 | 18,934.83 | Market Price |
| **Auxiliaries** | | | | | **5.58** | **(67,957.97)** | |
| | Foot insoles | 501 | 474 | 975 | 9.69 | 9,447.75 | Market Price |
| | Crutches | 377 | 331 | 708 | 16.26 | 11,512.08 | Market Price |
| | Wheelchairs | 111 | 148 | 259 | 181.46 | 46,998.14 | Market Price |
| **Total cost** | | 2,687,697.38 | 2,713,036.26 | | | **5,400,733.64** | |
| **Average cost** | | 400.19 | 497.53 | | | *443.81* | |

Note: average exchange rate for 2019; **ZAR14.4496: US$1.00**

prevention protocols in order to improve health outcome by mitigating LBP disability and its economic impact.

Outpatient costs had the highest costs, contributing about 38.9% (US$2.10 million) of the total costs. This was expected as outpatient care involves multiple visits for both ALBP and CLBP. This finding concurs with a study done in Netherlands by van Tulder who reported that the outpatient cost were US$2.1 million [14]. Expectedly, inpatient care had the second highest cost of US$2.08 million contributing about 38.5% of the total cost. However, van Tulder et al. reported that inpatient costs were higher than outpatient costs [ref]. This difference can also be attributed to differences in healthcare service delivery systems among countries such as accessibility, affordability and availability of services, and differences in study methodologies such as the method of costing (prevalence-based, incidence-based, human capital approach, friction cost or the willingness to pay method) and/or perspective of costing (societal, patients or providers perspective). Inpatient care includes costs for admission and the various professionals a patient interacts with during the hospital stay [29]. The high costs underscores the need for institution measures to sensitize and educate the public about LBP

**Table 5. Outpatient costs comparison between acute and chronic LBP.**

| Outpatient | Acute LBP | | CLBP | |
|---|---|---|---|---|
| | n = 9212 (75.70%) | | n = 2957 (24.30%) | |
| | Average total cost | % of Total costs | Average total cost | % of Total costs |
| Total visits | 100,924.74 | 1.8 | 84,856.50 | 1.5 |
| *Medication* | *760,294.08* | *13.6* | *920,346.96* | *16.5* |
| NSAIDs | 626,939.04 | 11.2 | 527,124.00 | 9.4 |
| Opioids | 133,355.04 | 2.4 | 202,895.28 | 3.6 |
| Antidepressants | - | - | 190,327.68 | 3.4 |
| **Rehabilitation** | 49,138.92 | **0.9** | 158,108.16 | **2.8** |
| Insoles | 5,581.44 | 0.1 | 3,856.62 | 0.1 |
| Total Cost | **915,939.54** | **16.4** | **1,009,060.08** | **18.0** |

prevention measures to limit cases of admission which comes with increased consumption of medication and significant disability [29].

Outpatient care costs were presented in both ALBP and CLBP. All cases report at the outpatient department for initial management before referrals for admissions or rehabilitations. It was noted that medication was the main cost driver across all the LBP sub-categories (ALBP and CLBP). The most commonly prescribed medication for LBP was NSAIDs. The total annual average cost of medication for ALBP and CLBP were comparably similar. This is potentially because the many cases of ALBP have less hospital visits while the few CLBP cases had numerous hospital visits. This finding is consistent with what was observed by Hong et al. in their cost of illness study in the UK [30]. Consumption of NSAIDS and opioids was noted to be frequent and indicated by the high costs. Due to the non-specific nature of LBP, pain medication is the most common treatment of convenience [25, 31] Interestingly, the costs increased with age. Again, this is because the prevalence of CLBP increases with age and is associated with multiple consultation, and or therapeutic interventions [30].

To our knowledge this is a first study to estimate costs of LBP in South Africa. The findings show the direct medical costs associated with LBP in primary care. Low back pain is a condition that has been reported frequently across population and under reported, yet its progress affect quality of life and can lead to loss of income due to disability and subsequently over consumption of medication [25]. Our findings indicate that LBP is of public health concern and should be prioritised as research has shown that the future predictions of its economic burden are substantial and continue to rise in low-and-middle-income-countries if no counteracting strategies are implemented [25]. As such it is imperative that LBP should form part of public health promotion and prevention messaging.

**Table 6. Per-patient average total cost for acute and chronic LBP.**

| Age | Acute LBP | Chronic LBP |
|---|---|---|
| | *Annual Per-Patient Average Total Cost* | *Annual Per-Patient Average Total Cost* |
| 18–27 | 60.92 | 1,107.29 |
| 28–37 | 66.23 | 737.05 |
| 38–47 | 79.31 | 1,349.11 |
| 48–57 | 102.89 | 1,822.25 |
| 58–67 | 136.70 | 1,840.97 |
| 68+ | 149.33 | 2,072.82 |
| **Mean** | **99.43** | **1,507.42** |

Whilst our study is presenting critical information on direct medical costs, we would acknowledge that our data was limited to only direct medical costs associated with outpatients, inpatient care, investigations, specialists, and use of auxiliary devices. In addition, results might not be representative at a national level because of the limited number of participating hospitals. However, the presented finds still suggest the need for action/attention toward recognizing LBP as one of the public health conditions needing attention and with great potential to have negative consequences on health resources. Secondly, it is likely that the reported numbers were underestimated. The ICD codes were handwritten, and this might have affected the reading and results in the exclusion of other potential files, therefore, we it is difficult to exclude selection bias.

## Conclusion

The direct medical expenditure for low back pain in KwaZulu-Natal is high mainly as a consequence of inpatient and outpatient care events. Outpatient care was the main cost driver and was significantly contributed by medication. Chronic LBP was responsible for the majority of costs, though it was represented by a small proportion of cases. The main cost drivers for CLBP were the inpatient care which involved ward admissions, nursing care and terminal care. Acute LBP only contributed a small percentage of the total costs, though it was represented by the majority of cases. The most common form of treatment for LBP was pain medication, of which NSAIDs was the most commonly prescribed medication, which was sometimes given in combination with opioids or antidepressants. Cost effective, culturally validated, context specific guidelines for the prevention of LBP should be developed and implemented. Measures to be taken to ensure practitioners and patients adherence to guidelines. Thus, this is important for policy makers, funders, stakeholders, and other involved actors to consider the prioritization of LBP research in the South African context to design cost-effective preventive measures. Urgent action should be taken to develop culturally validated guidelines based on local data to improve the future outcome of LBP and mitigate the burden thereof.

## Author Contributions

**Conceptualization:** Morris Kahere.

**Data curation:** Morris Kahere.

**Formal analysis:** Morris Kahere, Cebisile Ngcamphalala.

**Funding acquisition:** Morris Kahere.

**Investigation:** Morris Kahere.

**Methodology:** Morris Kahere, Cebisile Ngcamphalala.

**Project administration:** Morris Kahere.

**Resources:** Morris Kahere.

**Supervision:** Themba Ginindza.

**Validation:** Morris Kahere.

**Visualization:** Morris Kahere.

**Writing – original draft:** Morris Kahere.

**Writing – review & editing:** Morris Kahere, Cebisile Ngcamphalala, Ellinor Östensson.

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
