## [Decision Letter · Decision Letter 0]

30 Jun 2022

PONE-D-22-00974

The economic burden of low back pain in KwaZulu-Natal, South Africa: a prevalence-based cost-of-illness analysis from the healthcare provider’s perspective.

PLOS ONE

Dear Dr. Kahere,

Thank you for submitting your manuscript to PLOS ONE. After careful consideration, we feel that it has merit but does not fully meet PLOS ONE’s publication criteria as it currently stands. Therefore, we invite you to submit a revised version of the manuscript that addresses the points raised during the review process.

Please note that we have only been able to secure a single reviewer to assess your manuscript. We are issuing a decision on your manuscript at this point to prevent further delays in the evaluation of your manuscript. Please be aware that the editor who handles your revised manuscript might find it necessary to invite additional reviewers to assess this work once the revised manuscript is submitted. However, we will aim to proceed on the basis of this single review if possible. 

We look forward to receiving your revised manuscript.

Kind regards,

Steve Zimmerman, PhD

Associate Editor, PLOS ONE

**Journal requirements:**

a) Did participants provide their written or verbal informed consent to participate in this study?

“The authors would like to thank the University of KwaZulu-Natal (UKZN) for the provision of resources towards this project and the UKZN CHS Scholarship that was awarded to facilitate the research running costs.”

“This study was funded by the University of KwaZulu-Natal College of Health Sciences (CHS Scholarship). The funders had no role in study design, data collection and analysis, decision to publish, or preparation of the manuscript.”

6. Please remove your figures from within your manuscript file, leaving only the individual TIFF/EPS image files, uploaded separately.  These will be automatically included in the reviewers’ PDF.

7. Please include your tables as part of your main manuscript and remove the individual files. Please note that supplementary tables (should remain/ be uploaded) as separate ""supporting information"" files.

**Additional Editor Comments:**

Please change "females” or "males" to "women” or "men" as appropriate, when used as a noun (see for instance https://apastyle.apa.org/style-grammar-guidelines/bias-free-language/gender)."

Reviewers' comments:

Reviewer's Responses to Questions

**Comments to the Author**

1. Is the manuscript technically sound, and do the data support the conclusions?

Reviewer #1: Yes

2. Has the statistical analysis been performed appropriately and rigorously? 

Reviewer #1: Yes

3. Have the authors made all data underlying the findings in their manuscript fully available?

Reviewer #1: No

4. Is the manuscript presented in an intelligible fashion and written in standard English?

Reviewer #1: No

5. Review Comments to the Author

Reviewer #1: This is an interesting paper on the cost of LBP in a low income country. It is largely well written but does need a grammar and spell check. Specific comments are:

Abstract

1. Background: It may be better not to include production since this isn’t covered in the paper.

2. Results: Acronyms need to be defined at first use and used consistently - ALBP and CLBP

Background

3. P4 line 81: why is the pooled lifetime estimate lower than the 12 months estimate, shouldn’t it be the other way around?

4. Were there no prior South African studies or studies from other African countries to mention in the introduction?

5. Last paragraph: This should state what this study adds over prior studies. Line 279 page 11 in the discussion says this is the first study to estimate cost of LBP in South Africa. If so this should be stated in the last paragraph of the introduction.

Methods

6. P7 line 159 antidepressants are often used primarily to treat pain (often at a lower dose than for depression). Anticonvulsants are also commonly used in this way.

7. P 7 line 164 Were invasive procedures only in an emergency? Commonly surgery is recommended for chronic back pain but without an emergency situation or attendance at a hospital emergency department.

8. P 8 A paragraph at the end is needed to state ethics approval and software (and version) used to undertake the analysis (even if it is simply a spreadsheet such as Excel).

Results

9. P 9 line 227 The similar cost of ALBP and CLBP is worth mentioning in the discussion as in the introduction CLBP was significantly greater in the studies cited. It would be helpful to explain why the results of this study are different – for example if outpatient services are not easily accessible or are costly for the patient

Discussion

10. P10 lines 243 to 257 For each point, there needs to be a statement about what this current study found and how it compares with other studies. The differences with what is found in this study also need to be stated (e.g. ALBP and CLBP being similar is quite different to other studies that found expenditure on CLBP to be much greater than for ALBP).

11. P10 line 264: What were the differences in health care service delivery and study methodologies?

12. Line 279 page 11 in the discussion says this is the first study to estimate costs of LBP in South Africa. However lines 243-244 refers to several other studies of the economic burden on LBP in South Africa. This seems contradictory.

Grammatical/spelling errors

A grammar and spell check is needed. There are quite a few grammatical/spelling errors with a few examples being:

P 4 line 74 “Prevalence and incidence estimate” should be “The prevalence and incidence estimates”

p4 line 84 “…get completely healed” should be something like “do not make a full recovery”

P5 line 93 “lasted” should be “lasting”

P5 Line 110 “decision should be decisions”

P5 Line 111 ”hospital” should be ”hospitals”

P11 line 269 “All cases report at the outpatient ….” should this be “All cases report at the outpatient department…”?

6. PLOS authors have the option to publish the peer review history of their article (what does this mean?). If published, this will include your full peer review and any attached files.

Reviewer #1: **Yes: **Deborah Schofield

---

## [Author Response · Author response to Decision Letter 0]

16 Aug 2022

RESPONSE TO EDITORS’ COMMENTS

Response: we have addressed all formatting requirements, see the manuscript cover page

a) Did participants provide their written or verbal informed consent to participate in this study?

Response: This study was hospital chart review. There was no human participation, therefore no consent was required from participants. Authorisation to access the records was sought by means of gatekeeper permissions from participating hospitals.

“The authors would like to thank the University of KwaZulu-Natal (UKZN) for the provision of resources towards this project and the UKZN CHS Scholarship that was awarded to facilitate the research running costs.”

Response: All funding statements have been removed from the manuscript as suggested by the editor.

“This study was funded by the University of KwaZulu-Natal College of Health Sciences (CHS Scholarship). The funders had no role in study design, data collection and analysis, decision to publish, or preparation of the manuscript.”

Response: we have removed all the funding statements from the manuscript

Response: We have amended that statement, to state that, data for this manuscript is not available in any published paper/article, but raw data can be accessed upon signing of disclosure agreements between the interested parties and the university of KwaZulu-Natal. 

Response: We have moved the ethics statement up to the methods section and have deleted it from the declaration section. 

6. Please remove your figures from within your manuscript file, leaving only the individual TIFF/EPS image files, uploaded separately. These will be automatically included in the reviewers’ PDF.

Response: We have removed all figures from the manuscript and have uploaded them as separate files. 

Additional Editor Comments:

Please change "females” or "males" to "women” or "men" as appropriate, when used as a noun (see for instance https://apastyle.apa.org/style-grammar-guidelines/bias-free-language/gender)."

Response: We have changed “females” to “women” and “males” to “men”

RESPONSE TO REVIWERS COMMENTS

Reviewer #1: This is an interesting paper on the cost of LBP in a low-income country. It is largely well written but does need a grammar and spell check. Specific comments are:

Abstract

1. Background: It may be better not to include production since this isn’t covered in the paper.

Response: We have paraphrased the background section of the abstract to exclude the production cost as it does not concern the current research question, see page 2, lines 29 – 35 

2. Results: Acronyms need to be defined at first use and used consistently - ALBP and CLBP

Response: We have now defined all acronyms at first mention and are all used consistently now, see page 2, lines 48, 50, 54

Background

3. P4 line 81: why is the pooled lifetime estimate lower than the 12 months estimate, shouldn’t it be the other way around?

Response: This is the exact results of the study we gave reference of. You may find the article by following the DOI link: https://doi.org/10.1186/s12891-018-2075-x

4. Were there no prior South African studies or studies from other African countries to mention in the introduction?

Response: We have only identified one cos of illness study that was conducted in Africa, and we have now included it in the background section, see page 5, lines 116 – 128.

5. Last paragraph: This should state what this study adds over prior studies. Line 279 page 11 in the discussion says this is the first study to estimate cost of LBP in South Africa. If so, this should be stated in the last paragraph of the introduction.

Response: We thank the reviewer for this comment, and we have addressed that, see page 6, line 135

Methods

6. P7 line 159 antidepressants are often used primarily to treat pain (often at a lower dose than for depression). Anticonvulsants are also commonly used in this way.

Response: We thank the reviewer for the comment. We have now added that to the literature, see page7, lines 190 – 191 

7. P 7 line 164 Were invasive procedures only in an emergency? Commonly surgery is recommended for chronic back pain but without an emergency situation or attendance at a hospital emergency department.

Response: We thank the reviewer for the insight. We have addressed and rephrased that sentence, see page 7&8 lines, 194, 195

8. P 8 A paragraph at the end is needed to state ethics approval and software (and version) used to undertake the analysis (even if it is simply a spreadsheet such as Excel).

Response: We have deleted the ethical approval from the declaration section and inserted it as the last paragraph of the methods section, see page 8, lines 214 – 222 

Results

9. P 9 line 227 The similar cost of ALBP and CLBP is worth mentioning in the discussion as in the introduction CLBP was significantly greater in the studies cited. It would be helpful to explain why the results of this study are different – for example if outpatient services are not easily accessible or are costly for the patient

Response: The similarity between ALBP and CLBP was based on the fact that, there were many cases of ALBP with few presentations while the many cases of CLBP had several presentations over time, see page 12, lines 324 - 327.

Discussion

10. P10 lines 243 to 257 For each point, there needs to be a statement about what this current study found and how it compares with other studies. The differences with what is found in this study also need to be stated (e.g. ALBP and CLBP being similar is quite different to other studies that found expenditure on CLBP to be much greater than for ALBP).

Response: We have rephrased this section, see page 11, lines 228 – 303 

11. P10 line 264: What were the differences in health care service delivery and study methodologies?

Response: We have expanded on that statement to include the differences. The sentence now reads, “This difference can also be attributed to differences in healthcare service delivery systems among countries such as accessibility, affordability and availability of services, and differences in study methodologies such as the method of costing (prevalence-based, incidence-based, human capital approach, friction cost or the willingness to pay method) and/or perspective of costing (societal, patients or providers perspective)”, see page 11, lines 311 – 316 

12. Line 279 page 11 in the discussion says this is the first study to estimate costs of LBP in South Africa. However lines 243-244 refers to several other studies of the economic burden on LBP in South Africa. This seems contradictory.

Response: We have addressed that confusion. No study was conducted in South Africa, as this is the first one. The studies reported there are conducted elsewhere outside the African context, see page 11, lines 287 – 288 

Grammatical/spelling errors

A grammar and spell check is needed. There are quite a few grammatical/spelling errors with a few examples being:

P 4 line 74 “Prevalence and incidence estimate” should be “The prevalence and incidence estimates”

Response: We have changed “estimate” to “estimates”, see page 4, line 182

p4 line 84 “…get completely healed” should be something like “do not make a full recovery”

Response: We have changed “get completely healed” to “do not make full recovery”, see page 4, line 196

P5 line 93 “lasted” should be “lasting”

Response: We have changed “lasted” to “lasting”, see page 5, line 103

P5 Line 110 “decision should be decisions”

Response: “decision” has now been changed to “decisions”, see page 6, line 133

P5 Line 111 ”hospital” should be ”hospitals”

Response: “hospital” is now changed to “hospitals”, see page 6, line 134

P11 line 269 “All cases report at the outpatient ….” should this be “All cases report at the outpatient department…”?

Response: We have added, “department” as per the reviewers comment, see page 12, line 322

---

## [Decision Letter · Decision Letter 1]

20 Sep 2022

The economic burden of low back pain in KwaZulu-Natal, South Africa: a prevalence-based cost-of-illness analysis from the healthcare provider’s perspective.

PONE-D-22-00974R1

Dear Dr. Kahere,

We’re pleased to inform you that your manuscript has been judged scientifically suitable for publication and will be formally accepted for publication once it meets all outstanding technical requirements.

Kind regards,

Kuo-Cherh Huang

Academic Editor

PLOS ONE

Additional Editor Comments (optional):

Reviewers' comments:

Reviewer's Responses to Questions

**Comments to the Author**

1. If the authors have adequately addressed your comments raised in a previous round of review and you feel that this manuscript is now acceptable for publication, you may indicate that here to bypass the “Comments to the Author” section, enter your conflict of interest statement in the “Confidential to Editor” section, and submit your "Accept" recommendation.

Reviewer #1: All comments have been addressed

2. Is the manuscript technically sound, and do the data support the conclusions?

Reviewer #1: Yes

3. Has the statistical analysis been performed appropriately and rigorously? 

Reviewer #1: (No Response)

4. Have the authors made all data underlying the findings in their manuscript fully available?

Reviewer #1: No

5. Is the manuscript presented in an intelligible fashion and written in standard English?

Reviewer #1: Yes

6. Review Comments to the Author

Reviewer #1: All comments have been adequately responded to and I am happy for the paper for proceed to publication now

7. PLOS authors have the option to publish the peer review history of their article (what does this mean?). If published, this will include your full peer review and any attached files.

Reviewer #1: **Yes: **Deborah Schofield

---

## [Editor Report · Acceptance letter]

4 Oct 2022

PONE-D-22-00974R1 

The economic burden of low back pain in KwaZulu-Natal, South Africa: a prevalence-based cost-of-illness analysis from the healthcare provider’s perspective. 

Dear Dr. Kahere:

I'm pleased to inform you that your manuscript has been deemed suitable for publication in PLOS ONE. Congratulations! Your manuscript is now with our production department. 

Kind regards, 

on behalf of

Dr. Kuo-Cherh Huang 

Academic Editor

PLOS ONE